# Intestinal Microbiota-Derived Short Chain Fatty Acids in Host Health and Disease

**DOI:** 10.3390/nu14091977

**Published:** 2022-05-09

**Authors:** Jing Cong, Ping Zhou, Ruiyan Zhang

**Affiliations:** 1College of Marine Science and Biological Engineering, Qingdao University of Science and Technology, Qingdao 266042, China; 2Department of Oncology, Qingdao Central Hospital, Siliu South Road No.127, Qingdao 266042, China; zp880211@163.com (P.Z.); zhangruiyan88@163.com (R.Z.)

**Keywords:** short chain fatty acids, intestinal microbiota, dietary nutrition, effects, disease connection

## Abstract

Intestinal microbiota has its role as an important component of human physiology. It produces metabolites that module key functions to establish a symbiotic crosstalk with their host. Among them, short chain fatty acids (SCFAs), produced by intestinal bacteria during the fermentation of partially and non-digestible polysaccharides, play key roles in regulating colon physiology and changing intestinal environment. Recent research has found that SCFAs not only influence the signal transduction pathway in the gut, but they also reach tissues and organs outside of the gut, through their circulation in the blood. Growing evidence highlights the importance of SCFAs level in influencing health maintenance and disease development. SCFAs are probably involved in the management of host health in a complicated (positive or negative) way. Here, we review the current understanding of SCFAs effects on host physiology and discuss the potential prevention and therapeutics of SCFAs in a variety of disorders. It provides a systematic theoretical basis for the study of mechanisms and precise intake level of SCFAs to promote human health.

## 1. Introduction

The human intestine harbors trillions of bacteria, fungi, archaea, and viruses, which are often collectively referred to as “microbiota”. The intestinal microbiota has been considered as a fundamental player orchestrating host pathology and physiology [1]. Imbalance of intestinal microbiota is closely associated with diseases ranging from localized gastroenterological disorders to neurologic, respiratory, metabolic, hepatic, and cardiovascular illnesses [2]. Intestinal microbiota influences our health in multiple ways. An increasing body of evidence verifies that microbial metabolites have important effects on host physiology. Short chain fatty acids (SCFAs) are produced by intestinal microbes during the fermentation of partially and non-digestible polysaccharides. As the key player in the interplay between diet, microbiota, and health, SCFAs confer some degree of protection/destruction through a variety of pathways in a wide range of disorders. The purpose of this paper is to present a state-of-the-art review of our understanding of the role played by SCFAs in human health and disease and outline, as a result, possible further avenues of development of the field (Figure 1 and Figure 2). 

## 2. SCFAs—The What, Where, and How

SCFAs are important food ingredients with 2-6 carbon atom chains that are abundant in nature, mostly in the form of triglycerides in plant oils and milk [3]. Nevertheless, the main source of SCFAs in humans is microbial fermentation in the proximal intestines (jejunum, ileum, and proximal colon) and the distal intestines (descending colon, sigmoid and rectum) during the breakdown of undigested food fibers, as well as peptides and proteins [4,5,6]. Acetate, propionate, and butyrate account for more than 95% of SCFAs at an estimated ratio of about 3:1:1 in the gut [7,8]. The proportion can vary depending on diet, microbiota, host genotype, and site of fermentation [9]. These SCFAs range from 20–140 mM in the gut, with very high concentrations in the proximal colon (70–140 mM) and relatively low concentrations in the distal colon (20–70 mM) and in the distal ileum (20–40 mM) (Table 1) [7,10]. Propionate and butyrate are released in the distal intestines and are higher than in the proximal intestines. Acetate is released only in the distal intestines [6]. It could be ascribed to several reasons. One explanation is that it could metabolize a relatively higher proportion of SCFAs in the proximal intestinal mucosa [11]. Another explanation is that the composition and activity of intestinal microbiota differ from one bowel to the next, resulting in variable SCFAs production [12]. The capacity of intestinal epithelium cells (IECs) to absorb and transport SCFAs differs between the proximal and the distal intestines. It also suggests that SCFAs have a function in regulating health in the colon epithelium and immune cells on a local level [13]. Furthermore, the majority of SCFAs are utilized within the gut; however, a small proportion of SCFAs reaches the liver where can be used as substrates for energy-producing and metabolized to generate glucose.

High throughput sequencing technology has made great advances in characterization of microbes responsible for SCFAs production. SCFAs are mainly mediated by specific intestinal microbes from food (Table 2). Acetate producing pathways are widely distributed among microbial groups. For example, acetate can be produced by enteric bacteria and acetogens (*Blautia hydrogenotrophica*) via the acetyl-CoA and Wood-Ljungdahl pathways, respectively. However, pathways for propionate and butyrate are substrate specific and highly conserved. For example, propionate is mainly generated by two pathways: Firmicutes lactate pathway and Bacteroidetes succinate pathway [14]. Propionate production is dominated by relatively few bacterial genera. *Akkermansia municiphilla* has been identified as key propionate producing mucin degrading species. Butyrate is produced by Firmicutes, such as *Faecalibacterium prausnitzii*, *Eubacterium rectale*, *Eubacterium hallii*, and *Ruminococcus bromii* [15]. *Ruminococcus bromii* may ferment resistant starch to produce butyrate. Recently, there is emerging evidence that diet-driven changes in microbial diversity led to variations in SCFAs; however, these changes into long-term effects on host health need more intervention studies. In addition, changes in intestinal microbiota with patients have been linked with changed intestinal microbiota and a loss of SCFAs producing organisms. Therefore, it highlights that diet and intestinal microbiota have profound effects on SCFAs [16]. 

A fraction of SCFAs can directly across the epithelial barrier; however, most exists require specialized transporter for their uptake. Therefore, the majority of SCFAs across the mucosa need active transport delivered by two receptors: the monocarboxylate transporter 1 (MCT-1) and the sodium-coupled monocarboxylate transporter 1 (SMCT-1). MCT-1 and SMCT-1 are highly expressed on colonocytes and along the overall gastrointestinal tract. Acetate, propionate, and butyrate can be transported from the gut into lacteal lymphatic system and/or hepatic portal circulation, with total concentrations at about 79 μM, 148 μM, and 375 μM in peripheral, portal, and hepatic blood, respectively [7] (Table 1). Propionate and butyrate, metabolized by hepatocytes, have the concentrations at 1-15 μM in the systemic circulation, with acetate appearing between 100-200 μM [17] (Table 1). Acetate is either released into the peripheral venous system or remains in the liver [18]. As a result, only acetate could be detected in peripheral blood (Table 1). Butyrate, in particular, has more beneficial effects than other SCFAs, but it is difficult to deliver into the colon due to short half-life and low compliance rate with rectal administration [19]. It has the potential to locate and deliver butyrate-producing intestinal bacteria for solving this problem and improving our health.

**Table 2 nutrients-14-01977-t002:** The production, absorption, and transport receptors of major SCFAs.

SCFAs	Related Microbes	Absorption Site	G protein-Coupled Receptors
Acetate	Enteric bacteria and acetogens (*Blautia hydrogenotrophica*) [14]	Liver or peripheral venous system	GPR41, GPR43 [20,21]
Propionate	Firmicutes and Bacteroidetes [14]	Hepatocytes	GPR41, GPR43 [20,21]
Butyrate	*Faecalibacterium prausnitzii*, *Eubacterium rectale*, *Eubacterium hallii*, *Ruminococcus bromii* [15]	Colonocytes	GPR109A, GPR41, GPR43 [20,21,22]

## 3. SCFAs-Signaling Mechanisms

SCFAs have effects on biological responses based on two major signaling mechanisms (Figure 1). The first is that histone deacetylases (HDACs) influence gene expression by the direct inhibition, which is characteristic of propionate and butyrate. The second for SCFAs effects involves the signaling through G protein-coupled receptors (GPCRs), including GPR109A, GPR41, and GPR43 [23]. GPR43 expression has been identified along the overall gastrointestinal tract, including those cells of the immune system. In the immune system, GPR43 and GPR109A are expressed on neutrophils, macrophages, and dendritic cells (DCs), suggesting SCFAs’ roles in immune responses [24,25]. GPR43 with an anti-inflammatory role was also found in colitis and arthritis models [26]. GPR41 and GRP109A expressions activated by SCFAs were confirmed to have beneficial effects on body weight [27]. GPR43 and GPR41, sharing 43% amino acid identity, could bind to acetate, propionate, and butyrate [20] (Table 2). Among these SCFAs, butyrate is the most potent agonist for GPR109A, which is only expressed in humans, and propionate was more selective for GPR41 and GPR43 [22] (Table 2). There receptors could transmit signaling based on the different cell types and ligands. Therefore, we consider that it is valuable to develop certain drugs that target these receptors or their signaling pathways to influence SCFAs for disease treatment. 

## 4. SCFAs-Host Physiology

Usually, SCFAs may reduce proliferation and migration of immune cells, lower cytokine levels, and induce apoptosis to inhibit inflammation. However, marked changes of SCFAs concentrations in blood or tissues could cause inflammatory, immunological, and metabolic diseases [28]. Appropriate concentrations of SCFAs help to maintain normal metabolism in the prevention and treatment of disease [29]. Notably, due to insufficient daily dietary fiber intake and low abundance of butyrate-producing intestinal microbiota, SCFAs concentrations are probably less than optimal in older people [30]. Therefore, SCFAs are identified as important metabolic biomarkers of disease-related changes [31]. SCFAs not only play a role in intestinal homeostasis, but also are transported across the gut to enter the blood circulation and influence the function and metabolism of peripheral tissues. Recent research suggests that SCFAs have impacts in the disease etiology, correlating systemic effects with diet and intestinal microbiota. The current knowledge and therapeutic interest of SCFAs on some organs and tissues in and beyond the gut are outlined in the following text (Figure 2). Examples of SCFAs involved in various diseases in vivo and in vitro studies were listed in Table 3, which were obtained by searching for “(short chain fatty acid [Title/Abstract]) AND (Inflammatory bowel disease/cancer/obesity/diabetes/kidney/hypertension [Title/Abstract])” on PubMed from 2000 to 2020. 

### 4.1. Inflammatory Bowel Disease 

Intestinal dysbiosis is identified in individuals with inflammatory bowel disease (IBD), with a decreased number of SCFAs-producing bacteria and a lower concentration of SCFAs in feces [54]. It has been reported that European children are susceptible to develop IBD than African children, due to decreased microbial diversity and a lack of SCFAs production [55]. Patients with ulcerative colitis have the intestinal dysbiosis, characterized by a reduction in butyrate-producing bacteria *Eaecalibacterium prausnitizii* and *Roseburia hominis* [56]. The administration of probiotic bacteria producing SCFAs and SCFAs supplements have been shown to reduce the pro-inflammatory microenvironment in animal models of IBD and human colitis [57]. SCFAs protect against IBD-related intestinal inflammation by immune regulation and intestinal epithelial barrier maintenance. SCFAs could bind to GPR43 and GPR109A, which are important for regulation of intestinal immunity, to trigger the development of regulatory T cell (Treg) [37]. SCFAs could protect mice against colitis by modulating colonic Treg number and activity in a GPR43-dependent manner [58], because GPR43-deficient (Gpr43^(−/−)^) mice showed unresolving or exacerbated inflammation in models of colitis [26]. SCFAs could change pro-inflammatory cytokine production in IECs via enhancing nuclear factor-kappa B (NF-κB) activation in toll-like receptor (TLR) ligand-responses [59]. SCFAs could suppress intestinal stem cell proliferation, regulate tight junction protein expression, and promote crypt IECs differentiation, all of which can influence IBD by playing a direct anti-inflammatory role in IECs [32,33]. SCFAs, particularly butyrate, could enhance interleukin-22 (IL-22) production in CD4+T cells and ILCs to alleviate colitis in mice by activating GPR41 and inhibiting HDACs [34]. Although it has not been made a standard protocol due to patients’ compliance and partial restricted indications, the administration of SCFAs or prebiotics that facilitate SCFAs production is increasingly used as an adjuvant therapy to improve the efficacy of conventional treatment such as corticosteroid and 5-aminosalicylic acid therapy.

### 4.2. Cancer

SCFAs have the ability to directly influence colonocyte proliferation and differentiation to help to prevent colonic cancer [60]. Patients with colorectal cancer have lower level of butyrate-producing bacteria and SCFAs [61]. SCFAs influence colon cancer partially in a GRCR-dependent way. The expression of GPR43 has been shown to be decreased in many colon cancer patients. Butyrate and propionate could inhibit colon cancer cell growth and cause apoptosis by activating the GPR43 pathway [36]. Inflammation-induced colon carcinogenesis is accelerated when GPR109A level is low. Butyrate with GPR109A could stimulate colonic DCs and macrophages to boost Treg number by raising the production of IL-10 and Aldh1a [37]. The mice of Niacr1^−/−^, encoding GPR109A, are inclined to suffer from development of colon cancer [37]. Niacin, a pharmacological GPR109A agonist, suppresses colon cancer in a GPR109A-dependent manner [37]. Butyrate acts as a potent HDAC inhibitor in colon cancer cells, enabling cell cycle arrest, differentiation, and apoptosis at physiological concentrations [62]. 

Recently, a substantial impact of SCFAs in affecting immunotherapy in patients with non-small cell lung cancer (NSCLC) has been considered to be a promising strategy for personalized management by identifying early progressor and long responder patients [63]. According to preliminary findings, propionate and butyrate were shown to be significantly related with long-term beneficial effects in NSCLC patients treated with nivolumab in second-line treatment [63]. However, another research reported that high blood butyrate and propionate levels restrained the higher proportion of Treg (CD80/CD86/CD4) induced by anti-CTLA-4 in melanoma mice and patients with ipilimumab therapy [64]. 

### 4.3. Obesity 

Obesity and overweight are complex health concerns of multifactorial etiology that have a high risk of causing other chronic diseases, such as type 2 diabetes (T2D), insulin resistance, fatty liver disease, cardiovascular disease, and others [65]. Since the 1940s, the role of SCFAs in adiposity has been studied in human, as well as in vitro and in vivo animals [66,67]. Changes in the level of SCFAs produced by intestinal microbiota have contributed to the development of obesity. Several animal studies have indicated that SCFAs administration could reverse or reduce weight and adiposity growth [68,69]. In obese mice, treatment with sodium butyrate leads to weight loss by boosting energy expenditure and fat oxidation [42]. In the presence of continuous food intake and physical activity, oral administration of acetate, propionate, and butyrate to mice fed a high-fat diet results in body weight reduction and insulin sensitivity augment [39]. According to a study involving Mexican children, excess weight and obesity were shown to have lower concentrations of fecal propionate and butyrate than normal children [40]. In contrast, obesity individuals have been linked to higher fecal concentrations of SCFAs, particularly propionate, than lean individuals [8,41]. Another study found that overweight and obese people in the Netherlands had higher fecal SCFAs concentrations than lean ones, indicating that they had a higher microbial energy harvest [41]. These findings are debatable. One reason is that SCFAs produced by intestinal microbial fermentation account for 5-10% of daily energy intake [70]. Another reason is probably that fluctuations in SCFA concentrations in the feces are caused by a general dysbiosis in intestinal microbial community, leading to increased or decreased production or mucosal absorption.

### 4.4. Diabetes

SCFAs exhibit the correlative beneficial effects in glucose homeostasis. SCFAs have the capacity to enhance glucose-stimulated insulin secretion and control glucose based on the SCFAs-GPR axis [45,71]. Propionate could activate ileal mucosal GPR43 to reduce hepatic glucose production in healthy rats in vivo [44]. Propionate potentiated glucose-stimulated insulin release in vivo and in vitro studies [45]. Marino et al. found that non-obese diabetic (NOD) model mice, which were given a combination of acetate and butyrate by drinking water, had suffered less from type 1 diabetes (T1D) [43]. Moreover, acetate and butyrate improved intestinal barrier function and increased the number of *Bacteroides* species in NOD model mice, which helped to inhibit T1D [43]. Zhao et al. suggested that T2D could be a consequence of deficiency in SCFAs produced by carbohydrate fermentation in the gut [72]. However, researchers also discovered that mice exposed to more than 150 mM SCFAs developed urethritis and hydronephrosis, as well as alterations in lymphocytes [73]. The causes need further investigation. 

### 4.5. Kidney Diseases

Kidney diseases are partly developed by inflammatory processes, which have effects on the physiological response to renal infection and injury and contribute to the development of possibly irreversible kidney damage [74,75]. Here, acute kidney injury (ACI) and chronic kidney diseases (CDK) are the key topics. Growing evidence has highlighted that SCFAs play a favorable role in kidney diseases in both experimental animals and patients. Treating with acetate or acetate-producing bacteria could alleviate ACI in the ischemia-reperfusion injury model [46]. The cause could be the function of SCFAs in inhibiting inflammatory, reactive oxygen species, apoptosis, and chromatin modification [46]. Long-term oral administration of sodium butyrate attenuated ACI via improving the activity of renal antioxidant enzymes in an animal experimental model [47]. Increased dietary fiber significantly increased intestinal epithelial tight junctions, reduced oxidative stress and inflammation, and slowed severe renal dysfunction in CDK rats [76]. Furthermore, providing butyrate post-treatment to juvenile diabetic rats showed that SCFAs reduced renal histological alterations, DNA damage, and apoptosis in addition to lowering creatinine, plasma glucose, and urea [48]. However, there are still a number of unfavorable findings between SCFAs and CDK. A study found that oral dosages of SCFAs higher than physiological levels in mice produced Th1 and Th17 cells to induce inflammation and trigger kidney hydronephrosis [73]. Mice on a high-fiber diet had higher gut butyrate levels but were more susceptible to *E. coli* infection [49]. The underlying mechanism is still not clear. Furthermore, individuals with kidney diseases had the reduced bacteria that possessed SCFAs-forming enzymes compared to control groups [77]. Reduced dietary fiber intake was strongly associated with a reduction in the population of butyrate-forming bacteria in individuals with end-stage renal disease, and following dietary changes, the pathology ameliorated [77]. The effects of SCFAs on kidney diseases are mainly by decreasing inflammation and increasing antioxidant activity. However, SCFAs have only indirect favorable effects on remission of kidney diseases in human. These studies are either preliminary or controversial. More research is needed to fully comprehend the direct effects on kidney diseases, as well as better understand these mechanisms and their implications.

### 4.6. Hypertension

Hypertension is involved in a metabolic disorder. By activating T cells and macrophages with angiotensin II (AngII), hypertension could cause organ damage by infiltrating target organs such as the vasculature and the heart [78]. The organ damage caused by hypertension depends on the pro- and anti-inflammatory immune cells, as well as the hemodynamic load [52]. Intestinal microbiota and their metabolites SCFAs were supposed to be associated with hypertension [79]. *Lactobacillus* was decreased and Th17 cells were increased in a hypertensive mice model fed a high salt diet. After *Lactobacillus* supplementation, the number of Th17 cells and systolic blood pressure were both reduced [80]. SCFAs influence hypertension via interacting with GPCR receptors in the vascular tissues [51]. Propionate supplement caused the reduction of blood pressure in the Olfr78 gene knockout mice model, probably due to the GPR41 receptor activated in the vascular endothelium [51]. Bartolomaeus et al. discovered that propionate intervention could protect from hypertensive damage by mitigating systemic inflammation based on the reduction of Treg in wild-type NMRI and apolipoprotein E knockout-deficient mice model [52]. Butyrate supplement was reported to ameliorate endothelium-dependent vasodilation in a hypertension mouse model induced by chronic angiotensin II infusion [53]. Ganesh et al. identified that acetate was a key player in a hypertensive rat model of obstructive sleep apnea [50]. Clinical evidence supported the idea that microbially mediated SCFAs had anti-hypertensive properties, and that a high intake of fruits and vegetables lowered the incidence of hypertension [81].

### 4.7. Liver Diseases

The alleviation effects of SCFAs on liver diseases have been well established in both animal and human investigations. In animal studies, sucralose intake in maternal mice reduced SCFA-producing bacteria and diminished cecal butyrate production in offspring, causing hepatic steatosis to exacerbate in adulthood [82]. Both sodium acetate and sodium butyrate supplementation protected against nicotine-induced excess hepatic steatosis and western-style diet-induced non-alcoholic steatohepatitis (NASH) [83,84]. In human investigations, SCFAs were shown to be lower in non-obese non-alcoholic fatty liver disease (NAFLD) patients than in non-obese healthy people [85]. 

SCFAs contribute to prevent the development of NAFLD through a variety of mechanisms. The first is that SCFAs have key effects on fatty acid metabolism and visceral adipose tissue (VAT), both of which play a vital role in the development of NAFLD. Excess accumulation of VAT could increase the release of free fatty acids (FFAs) into liver [86]. The FFAs are considered as a key role in the development of NAFLD because they induce hepatic TNF expression via activating NF-κB [87]. In addition, VAT causes an imbalance of pro-inflammatory and anti-inflammatory adipokines, resulting in systematic inflammation, including the liver inflammation. In human multipotent adipose tissue-derived stem adipocytes, acetate could attenuate hormone-sensitive lipase (HSL) phosphorylation in a Gi-coupled way [88]. In rabbits, acetate could retrain lipid accumulation, increasing lipolysis and fatty acid oxidation while retraining fatty acid synthesis [89]. The second is that SCFAs could increase energy harvest, improve nutrient absorption, and enhance hepatic lipogenesis by regulating intestinal mobility. The primary mechanism is that activation of GPR41 and GPR43 promotes secretion of 5-hydroxytryptamine, peptide-YY, and glucagon-like peptide-1, which could impede intestinal transit and reduce gastric emptying, food intake, and intestinal motility [90]. The third is that SCFAs could protect the intestinal barrier. For example, acetate, propionate, and butyrate could stimulate intestinal Nod-like receptor family pyrin domain containing 3 (NLRP3) inflammasome to increase IL-18 secretion further to ameliorate intestinal barrier integrity [91]. The fourth is that SCFAs could come into the liver directly via the portal vein, inhibiting inflammation and hepatic steatosis. Acetate, propionate, and butyrate could relieve hepatic steatosis in the liver by activating AMP-activated protein kinase, expressing fatty acid oxidation gene, and blocking macrophage proinflammatory activation [92]. SCFAs also have a role in the epigenetic regulation of NAFLD development. Propionate, acetate, and butyrate as inhibitors of histone deacetylases, play a key role in NAFLD and prevent gene transcription by reducing histone-bound acetyl groups [93,94].

### 4.8. Immune System Diseases

SCFAs are involved in immune system diseases by modulating the production of immune mediators, cytokines, and chemokines, as well as regulating the differentiation, recruitment, and activation of immune cells, such as neutrophil, macrophages, DCs, and T lymphocytes [95]. Butyrate had direct salutary effects on target tissues of graft-versus-host disease (GVHD). Supplementing with butyrate could help to improve junctional integrity of IECs, reduce apoptosis, and mitigate the severity of GVHD [96,97]. Recently, enteral nutrition has been shown to have a higher advantage in terms of GVHD protection than parenteral nutrition because it favors a prompt recovery of the SCFAs-producing intestinal microbiota [98]. Propionate and butyrate could ameliorate bone destruction in rheumatoid arthritis, raise systemic bone density, modify the metabolic state of pre-osteoclasts, protect against postmenopausal bone loss, and mitigate arthritis [99]. Butyrate may increase Treg level in colon lamina propria and induce IL-10 secretion in DCs and macrophages to regulate immune functions, which helps to prevent the progression of multiple sclerosis [29]. Furthermore, SCFAs can also shape the immunological environment and influence the allergic inflammation. High levels of propionate in the blood protected against allergic inflammation in lungs by enhancing generation of macrophage and DC precursors depending on GPR41 [100]. 

### 4.9. Neurological/Psychiatric Diseases

SCFAs have the potential contributions in determining the etiology and treatment of stress-related disorders [101]. SCFAs supplementation in mice improved sociability, anxiety and depressive-like behaviors, cognition, and stress response [102]. Butyrate has been reported to ameliorate cognitive impairments in mid-adult high-fat-diet-induced obese mice model [103]. Propionate reduced anticipatory reward responses to high-energy food in the human striatum [104]. However, in a previous study, propionate-treated rats showed a series of autism spectrum disorders (ASD)-linked neurochemical changes, which was found to contribute directly or indirectly to acquired mitochondrial dysfunction in breaking carnitine-dependent pathways, similar to the finding in ASD patients [105]. Altogether, these findings suggest that SCFAs have a complicated regulatory mechanism in a variety of diseases. 

## 5. Conclusions

A continually emerging body of evidence supports the role of SCFAs by microbial fermentation in the colon as key mediators potentially aiding disease prevention, recovery, and slowing progression in animal models and preclinical trials. However, there are still some pathogenic/disease-causing consequences in response to diverse diseases due to the differences in SCFAs level, thus, an optimal health-promoting level of SCFAs is still an open question. Indeed, the variability in intestinal microbiota of different individuals exerts the overarching effects in SCFAs level. Additionally, considering that the amount and the variety of dietary partially and non-digestible polysaccharides ingested is difficult to determine whether it is a direct effect of SCFAs or intestinal microbiota that influence host homoeostasis. Therefore, future work remains to be done to comprehensively explore the mechanisms of SCFAs alone or with other metabolites in physiology and pathophysiology of the gut, as well as other tissues and organs in host. Furthermore, more SCFAs-related dietary strategies and pharmacological drugs might develop new possibilities for precision prevention and therapy of various diseases in the future.

## Figures and Tables

**Figure 1 nutrients-14-01977-f001:**
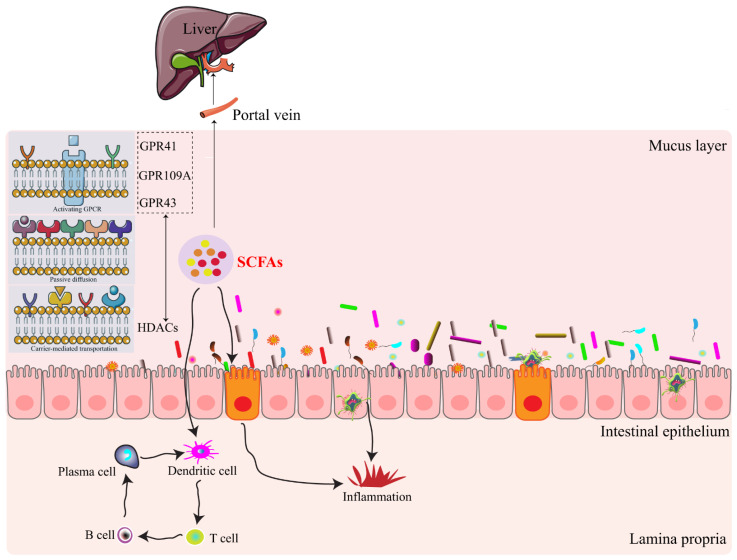
Interactions of short chain fatty acids with intestinal epithelium based on histone deacetylases (HDACs) and G protein-coupled receptors (GPCRs). Intestinal microbiota-derived metabolites of SCFAs are absorbed in direct or indirect ways. They are then absorbed into the blood vessels and finally enter the liver through the portal vein, as well as interact with the intestinal immune system. GPR41, G-protein coupled receptor 41; GPR43, G-protein coupled receptor 43; GPR109A, G-protein coupled receptor 109A.

**Figure 2 nutrients-14-01977-f002:**
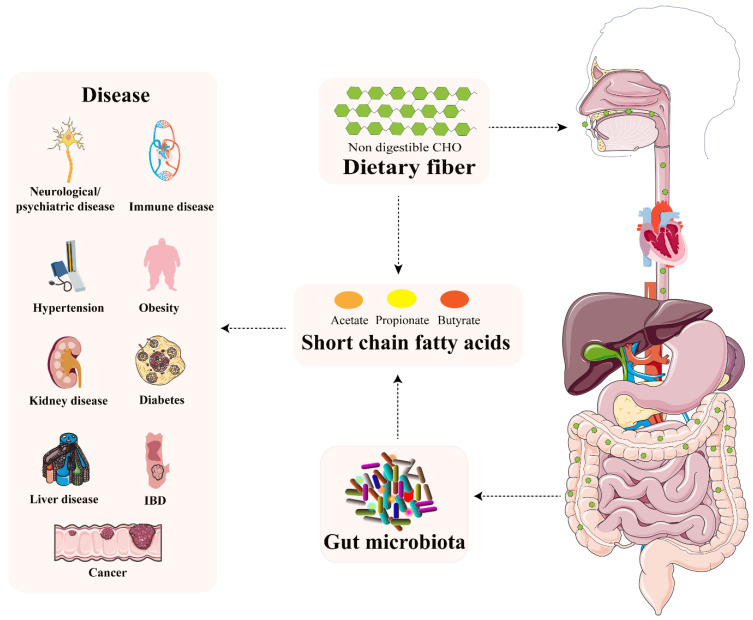
Intestinal microbiota-derived metabolites SCFAs result in the alteration of intestinal barrier function and are closely linked with host health and disease.

**Table 1 nutrients-14-01977-t001:** Gradient concentrations of SCFAs between gut lumen and periphery.

SCFAs	Proximal Colon	Portal Vein	Hepatic Vein	Systemic	Peripheral	Cerebrospinal Fluid
Acetate	42–84 mM	375 μM	148 μM	100–200 μM	79 μM	35 μM
Propionate	14–28 mM	-	-	1–15 μM	-	-
Butyrate	14–28 mM	-	-	1–15 μM	-	-

**Table 3 nutrients-14-01977-t003:** Examples of SCFAs involved in the diverse diseases in vivo and in vitro studies for recent 20 years (2000–2020).

Related Diseases	Relevant Short Chain Fatty Acids	Publications (2000–2020)
Inflammatory bowel disease	Butyrate [32,33,34,35]	174
Cancer	Propionate [36], Butyrate [37,38]	413
Obesity	Acetate [39], Propionate [8,39,40,41], Butyrate [39,40,42]	493
Diabetes	Acetate [43], Propionate [44,45], Butyrate [43]	367
Kidney diseases	Acetate [46]; Butyrate [47,48,49]	108
Hypertension	Acetate [50], Propionate [51,52], Butyrate [53]	79

Number of results obtained by searching for “(short chain fatty acid [Title/Abstract]) AND (Inflammatory bowel disease/cancer/obesity/diabetes/kidney/hypertension[Title/Abstract])” on PubMed from 2000 to 2020.

## Data Availability

Not applicable.

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
