# Peer review of "Intestinal Microbiota-Derived Short Chain Fatty Acids in Host Health and Disease"

_nutrients, 2022, doi:10.3390/nu14091977_

Round 1

Reviewer 1 Report

Thank you for giving me again the opportunity to revise the manuscript titled "Intestinal microbiota-derived short-chain fatty acids (SCFA) in host health and disease" Jing Cong and colleagues, have reported that SCFAs regulate colon physiology and change the intestinal environment, and there is increasing evidence that they play important roles in health maintenance and disease development. In this review, we will highlight the current understanding of the health effects of SCFAs as well as possible SCFAs therapeutics for a variety of disorders. SCFAs are probably involved in the management of host health in a complicated (positive or negative) way. Considering it, we should pay more attention to the precise intake levels of SCFAs for disease prevention and treatment. 

-Thanks again to the authors for incorporate my comments in the previous revisions. Reading this new version, seems that some numbers are missing in the titles and some letters could be more bigger, just minor commnets.

In general the review covers the main important information about SCFA, thank you to the authors for the figures. My only concern perhaps is a final paragraph or section as further perspectives.

Author Response

ResponseThank you for your suggestion and attention. We have revised the final paragraph like that “A continually emerging body of evidence supports the role of SCFAs by microbial fermentation in the colon as key mediators potentially aiding disease prevention, recovery, and slowing progression in animal models and preclinical trials. However, there are still some pathogenic/disease-causing consequences in response to diverse diseases due to the differences in SCFAs level, thus, an optimal health-promoting level of SCFAs is still open question. Indeed, the variability in intestinal microbiota of different individuals exerts the overarching effects in SCFAs level. Additionally, considering that the amount and the variety of dietary partially and non-digestible polysaccharides ingested is difficult to determine whether it is a direct effect of SCFAs or intestinal microbiota that influence host homoeostasis. Therefore, future work remains to be done to explore comprehensively the mechanisms of SCFAs alone or with other metabolites in physiology and pathophysiology of the gut, as well as other tissues and organs in host. Furthermore, more SCFAs-related dietary strategies and pharmacological drugs might develop new possibilities for precision prevention and therapy of various diseases in the future”.

Reviewer 2 Report

This manuscript by Dr. Jing Cong et al reviewed the effect of Intestinal microbiota-derived short chain fatty acids in host health and disease, and concluded that the importance of SCFAs level in influencing health maintenance and disease development. It is generally good, but has many typoes.

Minor comments:

  1. Line #379. HDAC (singular).
  2. Line #380. Delete attenuate.
  3. Line #386. Nod like (delete e).

Author Response

Response: Thank you for your suggestion. There are some problems about typos when we used the template. We have revised it in new version according to the journal template.

Line 379, we have revised the singular “HDAC” in line 379 of our manuscript.

Line 380, we have deleted it.

Line 386, we have deleted it.
